# Flame Retardant Behaviour and Physical-Mechanical Properties of Polymer Synergistic Systems in Rigid Polyurethane Foams

**DOI:** 10.3390/polym14214616

**Published:** 2022-10-31

**Authors:** Branka Mušič, Nataša Knez, Janez Bernard

**Affiliations:** Slovenian National Building and Civil Engineering Institute, 1000 Ljubljana, Slovenia

**Keywords:** flammability, polyurethane polymer, foams, thermal conductivity, mechanical properties

## Abstract

In the presented work, the influence of two flame retardants—ammonium polyphosphates and 2,4,6-triamino-1,3,5-triazine on the polyurethane foam (PUR) systems were studied. In this paper, these interactive properties are studied by using the thermal analytical techniques, TGA and DTA, which enable the various thermal transitions and associated volatilization to be studied and enable the connection of the results with thermal and mechanical analysis, as are thermal conductivities, compression and bending behavior, hardness, flammability, and surface morphology. In this way, a greater understanding of what the addition of fire retardants to polyurethane foams means for system flammability itself and, on the other hand, how this addition affects the mechanical properties of PUR may be investigated. It was obtained that retardants significantly increase the fire resistance of the PURs systems while they do not affect the thermal conductivity and only slightly decrease the mechanical properties of the systems. Therefore, the presented systems seem to be applicable as thermal insulation where low heat conductivity coupled with high flame resistance is required.

## 1. Introduction

Polyurethane foams are materials known for a long time, the use of which continues to increase over the years. In addition to the already known wide applicability in various industries such as construction, e.g., for thermal insulation in windows and doors and fastening and sealing of joinery, in the automotive industry, as well as in households for various purposes, for fastening fence posts in the garden and as an electrical insulator, etc., recently it is even used in geotechnical applications for soil reinforcement [1] and as an insulating material for building walls, e.g., in attics. With increasing use, there is also increasing interest in potential improvements of this material and, consequently, also interest in the influence of various additives, especially in terms of reaction to fire and on the physical-mechanical properties of polyurethanes. Polyurethane foams are materials with a low weight-to-strength ratio, low electrical conductivity as well as low heat conductivity [2]. On the other hand, the downside of PURs is their high flammability [3].

Due to increased climate and environmental concerns, there was a need to design a new effective fire-retardant system from halogen-free fire retardants [4,5,6,7], with aluminum hydroxide (ATH), magnesium hydroxide (MDH), carbon nanotube (CNT), expandable graphite, halloysite nanotubes with POSS, etc. Furthermore, in the construction industry, a lot of effort has been put to fulfil ever stricter reactions to fire requirements, and therefore over the years a lot of knowledge has been accumulated about the influence of various environmentally acceptable additives and retardants to fire resistance of the PURs, with a synergistic effect on improving the thermal insulating properties [8,9]. However, there is less known about the influences of those fire-retardant additives on the mechanical properties of the PURs.

In the presented study, the influence of two compounds: the ammonium polyphosphates and 2,4,6-Triamino-1,3,5-triazine on fire resistance, thermal conductivity, and different mechanical properties of PURs systems were systematically investigated.

## 2. Materials and Methods

### 2.1. Materials

A two-component polyurethane foam “Tekapur Polefix“ (PURs), TKK d.o.o. (Srpenica, Slovenia) Component A is a polyol with several hydroxyl groups and triethyl phosphate. Component B is a polymethylene polyphenyl polyisocyanate.Ammonium polyphosphate, Exolite AP 422 (APP) was supplied by Clariant (Mutenz, Switzerland), it is Ammonium polyphosphate, white fine powder, non-hygroscopic, non-flammable, halogen-free, with bulk density 700 kg/m^3^, and melting point ~240 °C (decomposition).2,4,6-Triamino-1,3,5-triazine (TATA), Sigma Aldrich (St. Louis, MO, USA), white powder, with bulk density 800 kg/m^3^, and melting point ~354 °C (decomposition).

### 2.2. Preparation of PURs

All PURs were individually prepared according to the same procedure and using a mold in which PUR foam expanded. First, the appropriate amount of PUR component B and a flame retardant (except for PUR 0) were weighted into a mixing vessel and mixed with high-speed mechanical stirrer, at about 1400 rpm for 10 min to obtain a homogeneous mixture. After that, the appropriate amount of component A was poured into the mixture which was further homogenized with a stirrer at 1000 rpm and transferred into a mold with enough free space to enable the full expansion of the foam during curing. After about 45 s, the foam begins to expand. The foam reaches the final volume in about three minutes and after 10–15 min. The foams were allowed to cure for 72 h, at room conditions T = 23 ± 2 °C and relative humidity 50 ± 15 % in accordance with ISO 291:2008. After curing, the foams were cut into standard shaped specimens for further testing. When preparing the samples, we made sure that the samples were as uniform as possible.

The structure and resulting performance of polyurethane foams are driven by the stoichiometry of the polymerization reaction, which is directly impacted by applied monomers, additives, their chemical composition, and the ratio between the polyols and isocyanates [10]. The amount of hydroxyl and isocyanate groups present in the system are essential for reactions leading to the generation of urethane bonds [11,12].

The reference PUR without additions was designated as PUR 0; the foam with addition of APP was designated as PUR 1 and finally, the foam with the addition of TATA was designated as PUR 2. The ratio used in PUR 0 between polyol and isocyanate was according to the manufacturer’s recommendations, therefore the mixing weight ratio was 1:1.22. From preliminary research, we found that a maximum of 30% of the fire-retardant additive can be included in the system, based on the total weight of the A + B component, so that the fire retardant powder is homogeneously mixed into the B component, the expansion takes place on the scale of PUR 0 and the polymerization reaction ends (mass is not sticky after expansion). In Table 1 the contents of raw materials in PURs are given.

### 2.3. Methods of Characterization

Unless stated otherwise, before characterization the specimens were conditioned for at least 24 h at standard laboratory conditions at 23 ± 2 °C and 50 ± 5% relative humidity. Further on, mechanical properties and the apparent densities were determined at stated conditions also, as required by relevant standards. The published mechanical properties and apparent densities are presented as the average of the 5 measurements ± standard deviations, while other characteristics were obtained on single specimen measurement.

#### 2.3.1. Apparent Densities

The apparent densities of the PUR specimens were determined according to ISO 845:2006. The dimensions of the specimens were (50 mm × 50 mm × 50 mm) ± 1 mm.

#### 2.3.2. Thermal Conductivity

The thermal conductivity of the PUR specimens was determined in a home-made heat flow setup. Prior to testing the specimens were conditioned at 70 °C for 14 days and further two days at 23 °C, 50% RH. The dimensions of the specimens were (100 mm × 60 mm × 10 mm) ± 1 mm. Thermal conductivity was determined on the specimens inserted in-between cold and hot plates with temperatures of 15 °C and 25 °C, respectively.

#### 2.3.3. Thermal Decomposition

Thermal decomposition of the PUR specimens was determined with thermogravimetric analysis (TG), Netzsch instrument STA 409PC Luxx, Weyhe, Germany. The specimens with a mass of about 25 mg were heated in airflow from room temperature to 900 °C with rate of 10 K/min.

#### 2.3.4. Compression and Bending Behavior

The compression and bending behavior of the PUR specimens were determined on a universal test machine Zwick Z030, Zwick Roell Group, Ulm, Germany. Compression properties were determined according to EN 826:2013. The test specimens of dimensions (50 mm × 50 mm × 50 mm) ± 1 mm were compressed between the two plates of the universal test machine and at a constant rate of 0.5 mm/min was applied to the specimen till failure occurred. Bending behavior was determined according to the requirements of EN12089:2013. The specimens of dimensions of (150 mm × 30 mm × 50 mm) ± 1 mm were tested in three-point bending mode in a universal test machine. At a constant rate of 0.5 mm/min till failure occurred.

#### 2.3.5. Hardness

The hardness of the samples was measured using a device known as a Durometer and the determined hardness values are therefore referred to as durometer hardness. Durometer hardness is a dimensionless quantity; it represents a relative comparison of hardness between different, yet similar grades of materials, having hardness measured on the same durometer scale. The Shore A hardness tester (Zwick, Ulm, Germany) was used for determining the hardness of PUR samples, according to EN ISO 868:2004. For each sample, eight measurements were taken.

#### 2.3.6. Flammability

Flammability of the PURs were obtained according to UL-94 HB on the specimens with dimensions of 125 mm × 15 mm × 100 mm) ± 1 mm. A Horizontal burning test was performed.

#### 2.3.7. Cone Calorimetry

Reactions to fire properties were studied by using a cone calorimeter, produced by Fire Testing Technology, East Grinstead, UK according to ISO 5660-1:2015. Specimens were exposed to a heat flux of the 40 kW/m^2^.

#### 2.3.8. Loss of Ignition Test (LOI)

Loss on ignition was assessed from the weight of the test specimens before and after the exposure to the 40 kW/m^2^ in a cone calorimeter.

#### 2.3.9. FTIR Analysis

Exhaust gases released during exposure of the test specimens to the 40 kW/m^2^ in the cone calorimeter were analyzed by means of FTIR analyzer atmosFIR produced by Protea, Middlewich, UK according to ISO 19702:2015.

#### 2.3.10. Surface Morphology

The distribution of solid flame-retardant particles and the shape and size of sample porosity was observed using a scanning electron microscope (SEM) JSM-IT500LV, Oxford Inca; Jeol, Oxford Instruments Analytical (Freising, Germany), with an integrated energy-dispersive spectroscopy, W filament, fully automatic gun alignment, and in low (10–650 Pa) vacuum mode.

## 3. Results

### 3.1. Apparent Densities

The densities of the PURs are as follows PUR 0 (46.70 kg/m^3^), PUR 1 (62.61 kg/m^3^), and PUR 2 (60.04 kg/m^3^). The densities of PUR 1 and PUR 2 are comparable and about 30% higher than PUR 0. For PUR 1 and PUR 2, the same amount of additive 25 mass % was added.

### 3.2. Thermal Conductivity

The thermal conductivity of the specimens are as follows PUR 0 (36.5 mW/mK), PUR 1 (36.4 mW/mK), PUR 2 (35.6 mW/mK). Presented values correspond well to apparent densities of the specimens as a higher density of the cellular insulation generally contributes to increasing its thermal conductivity.

### 3.3. Thermal Decomposition

Thermal decomposition curves (TG) are presented in Figure 1. The mass losses at the first decomposition step were 36.3 wt.% for PUR 0, 28.3 wt.% for PUR 1, and 51.5 wt.% for PUR 2. Respectively, the mass losses for the second step were 55.6 wt.%, 65.4 wt.%, and 46.0 wt.%. Decomposition steps end at 338 °C and 662 °C for PUR 0, at 293 °C and 815 °C for PUR 1 and at 336 °C and 638 °C for PUR 2. TG curves of PUR specimens are presented in a Figure 1.

The first decomposition steps of all the PUR specimens were completed in a relatively narrow temperature range in-between 293 °C and 338 °C.

From Figure 1 we can see also that the course of the weight loss curve is similar for PUR 0 and PUR 2, while different for PUR 1. The weight loss is slightly lower at PUR 1 and PUR 2 than at PUR 0 up to a temperature of 293 °C. In PUR 1 was added APP, a high molecular weight phosphate-based chain, it serves as both an acid source and a blowing agent in intumescent formulations known to promote char formation during polymer decomposition. At elevated temperatures, the phosphorus containing the flame-retardant additive, APP, decomposes to produce phosphoric and polyphosphoric acids, which consequently promote charring via cross-linking of reactive polymer fragments [13]. The formation of carbonized char networks prevents or slows the transfer of heat, oxygen, and combustible volatiles into the pyrolysis zone; hence retarding the flaming/combustion process. Detailed mechanistic schemes describing the charring behavior of APP containing resin formulations have been discussed by Kandola and Ullah [13,14]. Values of partial weight loss in sample PUR 2 in lower temperature ranges are related to water evaporation. Weight loss at around 336 °C in PUR 2 was also due to partial loss of formaldehyde, methanol, and amine. The polycondensation reaction of melamine took place at temperatures above 336 °C when the products underwent a number of independent reactions involving both side chain and ring degradation. This means that some melamine molecules can be sublimated at a temperature lower than the sublimation temperature typically observed at 345 °C. Weight loss also occurs due to the release of formaldehyde, methanol, amine, and NH_3_ from melamine (at about 390 °C). Weight loss at temperatures above 450 °C involves the general thermal decomposition of melamine, which ends above 660 °C with the decomposition of melamine to form volatile products, including CO_2_, HCN, and CO [15].

### 3.4. Compression and Bending Behaviour

The compressive properties were determined on three parallel samples. Figure 2 show how the deformation of PUR 0, PUR 1 and PUR2 varied continuously with increasing standard force. The PUR 0 and PUR 2 samples behave similarly, while the PUR 1 sample has slightly worse result.

The bending properties were determined on three parallel samples. Figure 3 show how the deformation of PUR 0, PUR 1 and PUR2 varied continuously with increasing standard force.

The compressive and bending properties of the specimens are summarized and presented in Table 2. The (σ_M_) represents compressive strength and (σ_b_) bending strength.

### 3.5. Hardness

The Shore A scale is employed for softer/flexible materials. The measured values indicate the resistance to indentation of the tested material on a scale between 0 and 100.

The hardness test is based on the measurement of the penetration of a rigid peak into the specimen under specified conditions. The measured penetration is converted into International Rubber Hardness Degrees (IRHD). The hardness scale of degrees is chosen such that 0 represents a material having an elastic modulus of zero, and 100 represents a material of infinite elastic modulus.

Table 3 shows the durometer hardness of PURs samples, whereas a thumb rule, higher numbers on the scale indicate a greater resistance to indentation, which means harder material.

### 3.6. Flammability

Samples PUR 0, PUR 1, and PUR 2 were prepared and tested for combustion in accordance with the UL-94 HB standard. The results are shown in Table 4. The dripping of samples during burning did not occur in any case. The samples stopped burning immediately after removing the fire source. The fire reached the marked line of the PUR 0 after burning 30 s, whereas PUR 1 and PUR 2 preserved more unburning material than neat PUR 0. It can be observed that in the case PUR 1 and PUR 2 promoted the formation of a compact burned layer.

Visual differences between PURs, according to the UL-94 HB burning test, are presented for one set in Figure 4. We can see that in PUR 0 the line is no longer visible, in the case of PUR 1 with the fire-retardant additive APP the burning has reached the line, and in the case of PUR 2 the flame retardant TATA works even better.

Cellular material PUR 0 burns readily in the presence of oxygen and heat with a very high fire spread rate and a high smoke release rate. As per the experimental evidence, the pores of the foam entrap air further aid in its combustion [16,17].

The mechanism of APP (added to PUR 1) degradation has been investigated and consists of the release of water and ammonia and the formation of polyphosphoric acid, which is then volatilized and dehydrated at temperatures above 250 °C [18]. APP is also thought to promote an intumescent layer of char, which acts as a physical barrier to slow the mass transfer of heat. Due to both processes, the halogen-free flame-retardant APP is considered a very effective phosphorus-based flame retardant used in polymers because it is more environmentally friendly, highly effective, and low in toxicity. However, it is necessary to note, as can be seen from the results, that APP can affect the deterioration of the physical and mechanical properties of the composite and increase the generation of smoke [19,20,21,22].

In the PUR 2 sample, the flame-retardant melamine captures the heat of the PUR matrix during combustion and undergoes advanced endothermic condensation with the evolution of ammonia. In the first phase, water from the sample evaporates. This is followed by the breaking of urethane bonds and decomposition. Melamine does not begin to decompose until somewhere above 450 °C and involves general thermal decomposition of melamine, which ends above 660 °C [16]. According to the UL-94 HB test, we can see that PUR 2 had the best results.

### 3.7. Cone Calorimetry

Cubes of 50 mm made of the three PURs were cut in 10 mm thick squares. From each type of PUR two 100 mm × 100 mm, 10 mm thick specimens were prepared, two of each type of PUR. Specimens were exposed to the 40 kW/m^2^ heat flux in a cone calorimeter. For each type of PUR, a self-ignition of the exposed specimen was observed as well as ignition of the exposed specimen initiated by sparks. The heat release rate and total heat release parameters were compared for the three PURs under both conditions—without or with a help of a spark igniter [23,24,25,26]. Table 5 shows the appearance of PUR 0, PUR 1 and PUR 2 samples before, during and after the test.

From Figure 5 and Figure 6 we can see that all specimens ignited in a few seconds after the heat flux exposure. For PUR 1 and PUR 2 specimens a white smoke has been noticed before ignition. PUR 1 and PUR 2 specimens expanded as seen in the photo. After the test of PUR 0, only a very small number of residuals were left whereas PUR 1 and PUR 2 specimens still had a firm structure.

During heat flux exposure in the cone calorimeter, all specimens ignited in both ignition modes. From Table 6 we can see that the specimens ignited faster when a spark igniter was used.

The greatest difference in ignition time was for PUR 0. PUR 1 self-ignited fast, but the heat release rate was significantly lower compared to HRR when the ignition was induced with the spark igniter. The heat release rate for PUR 2 specimen was similar for both modes of ignition, for self-ignition and ignition with a spark igniter. In both PUR 2 specimens, the HRR curve has two peaks. In addition, THR is similar for the two ignition modes for PUR 2 whereas for PUR 0 and PUR 1 the THR is significantly lower when specimens were self-ignited.

The smoke production rate (Figure 7 and Figure 8) was similar for both ignition modes in all three PUR types. Smoke production was the greatest for PUR 0 specimens and the lowest for PUR 2 specimens [27]. Two peaks were noticed in PUR 2 specimens.

### 3.8. Loss of Ignition Test (LOI)

During exposure of the three PUR specimens to the 40 kW/m^2^ in a cone calorimeter, mass loss of the specimens was measured. We can see the mass loss shown in Figure 9. Two modes of ignition were compared for the three PURs—self-ignition and ignition with sparks. The difference between the mass before the test and the final mass after heat flux exposure was calculated for each PUR type and ignition mode. Loss on ignition was calculated as a percentage of mass loss compared to initial mass.

Loss on ignition was similar for the two ignition modes for all three PURs. On the other hand, LOI differs between different PURs, in PUR 0 LOI was around 95%, in PUR 1 around 65%, and in PUR 2 around 80%, as shown in Table 7.

### 3.9. FTIR Analysis

During heat flux exposure of test specimens in a cone calorimeter, the exhaust gases were continuously analyzed by an FTIR analyzer. Concentrations of several gases were calculated from IR spectra [28]. Such calculations can lead to certain inaccuracy, especially where the amount of certain gas is low or other gases with a similar spectrum are present. Negative values on graphs are the result of characteristics of the calculation method.

Concentrations of several gases were calculated. Concentrations of carbon dioxide (Figure 10) and carbon monoxide (Figure 11) were measured with a cone calorimeter’s gas analyzer. A comparison of gas concentrations for the three PURs under both ignition modes was made.

During flaming both CO_2_ and CO were released. After the flame was extinguished, the concentration of CO increased significantly for PUR 0.

In addition, NO was released during flaming in all tested specimens, while NO_2_ was significantly noticed in PUR 0 and PUR 1 self-ignited specimens, which is clearly visible in Figure 12 and Figure 13.

We also detected the release of ammonia, which can be seen in Figure 14.

During exposure of the PUR 1 specimen to heat flux without sparks, some gases were detected, namely NH_3_, C_2_H_4_, and C_2_H_6_, which were not seen in other specimens. Only in PUR 0 specimen, ignited with a spark, was C_2_H_4_ released also. When comparing self-ignited PUR 1 with other specimens it was noticed that the heat release of that specimen was significantly lower than with other specimens. It is possible that the flow of the chemical reaction was different. For finding the cause, some further investigations would be needed.

### 3.10. Surface Morphology

Monitoring the integration of the fire-retardant powder and the porosity of the materials was key in the SEM analysis [29,30]. In micrographs (Figure 15, Figure 16 and Figure 17) of material surfaces with included fire-retardant powders are shown. The images were taken before (Figure 15) and after the addition of the fire-retardant powder (Figure 16 and Figure 17) at 30×, 50×, and 100× magnification.

It can be seen from Figure 15, Figure 16 and Figure 17 that PUR 0, without the addition of fire-retardant powder, forms the polymer network with the largest pores. In Figure 16 we can see in all three PUR 1 micrograms (a–c) that at the time of formation of the polymer network of the three-component PUR 1 composite, smaller pores were formed, which can be attributed to the addition of APP, and similarly, in Figure 17, where TATA was added as a fire-retardant powder, we can observe in PUR 2 micrograms (a–c) that even smaller pores were formed. The pores in all three prepared samples PUR 0–PUR 2 were random but evenly distributed. Figure 16 also shows many more solid particles on the surface than can be seen in Figure 15 and Figure 17. We assume that the particles in Figure 16, which are pure two-component resin (binder + hardener) without additives, are possibly residues of an unreacted component or impurities left after sample preparation/cutting for SEM analysis. However, the significantly higher number of solid particles in Figure 16 may also be attributed to the fact that the APP may be more difficult to mix with the selected resin system, thus making the inhomogeneity of the PUR 1 sample much worse. Evaluation analysis of the agglomerate formation of micrographs was not performed because the powder is integrated also within the cavities of foamed materials, which made evaluation unreliable.

## 4. Discussion

As a thermal insulation material, PURs shall meet the demand of flame resistance, and at the same time also need to possess the necessary physical-mechanical properties. In this work, the influences of fire retardants of ammonium polyphosphates and 2,4,6-triamino-1,3,5-triazine on the properties of 2-component polyurethane foam were investigated.

Even though the additions of flame-retardant specimens contribute to increasing the density of PUR 1 and PUR 2 specimens their compressive strengths and corresponding strains were lower as compared to the PUR 0 reference. In the case of PUR 1, a 33% decrease in compressive strength and an 11% decrease in bending strength were recorded. For PUR 2 about a 7% decrease in compressive and a 14% in bending strengths were observed. It can be concluded that retardants are not chemically bonded into PU binders. However, it is worth pointing out that mechanical properties are not considerably affected by the additions of retardants. Thus, PUR 1 and PUR 2 still exhibited relatively high mechanical properties as compared to typical cellular insulation to be used for thermal insulation of the buildings.

The shore durometer hardness value itself does not provide direct information on, e.g., strength or resistance to scratches, abrasion, or wear. This hardness is a measure of a material’s resistance to localize the plastic deformation, and it can be defined as a measure of a material’s resistance towards an external force applied to the material. From the results, we can see that the addition of fire-resistant powders influenced resin network forming and thus decrease the durometer hardness values, even by 50%.

It was determined that presented retardants considerably decrease the flammability of the systems, while thermal conductivities are not affected.

The fire behavior of different PUR specimens depends on fire conditions. In conditions of starting a fire as in UL 94 standard, additives in PUR successfully reduce fire spread. In conditions toward the fully developed fire as simulated with the cone calorimeter test, specimens react differently. Specimens with additives compared to PUR 0 specimens burn longer, the heat release rate is lower, but total heat release is higher. The smoke production rate for specimens with additives is lower compared to PUR 0, also total smoke production is lower, especially for PUR 2 specimens.

When comparing loss on ignition, PUR 0 specimens had the highest LOI value, around 95%. PUR 1 specimens had the lowest LOI value, around 65%.

When observing the release of gases during heat exposure, it was noticed that CO significantly increased after the end of flaming at PUR 0 specimens, while in other specimens CO was not observed after the end of flaming. NO_2_ and C_2_H_4_ were observed in PUR 0 self-ignited specimen and PUR 1 specimen, ignited by sparks. NH_3_ and C_2_H_6_ were observed only at heat exposure of PUR 1 specimen, when self-ignited. The different gas releases can indicate different chemical reactions and can relate to the low heat release rate of that specimen during heat exposure. For a better understanding of the chemical processes, it would be necessary to do new, in-depth research related to the type of test, the variations of the test parameters, the amount of fire-retardant additives, etc.

In addition, it was found that the foamed PUR materials without and with integrated different fire-retardant powders have different shapes and sizes of porous or polymeric net structures.

The above research was carried out based on three different composites, and we should mention that we did preliminary research with different amounts of fire-retardant material additives, i.e., 10% and 30%, and 50%. At 10% addition, there were no significant differences in the UL94 HB test, while at 50% addition, we failed to homogeneously mix such a large amount of dust into the B component, so we selected samples with 30% addition and re-prepared the entire batch of 30% samples for conditioning and further analysis. The present work is the basis for a further, more detailed study of polyurethane systems, in which additional materials affecting the response to fire will be investigated, different concentrations of additives or different compositions of the systems will be investigated, as well as different methods of installation procedures and the effect on their mechanical properties.

## Figures and Tables

**Figure 1 polymers-14-04616-f001:**
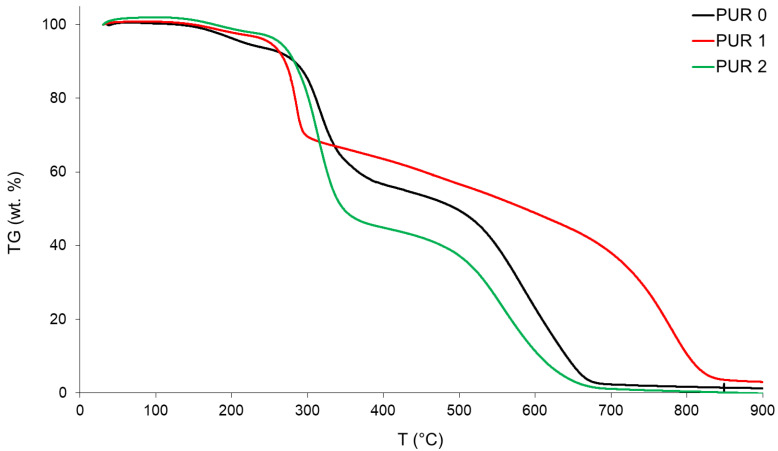
TG curves of the PURs.

**Figure 2 polymers-14-04616-f002:**
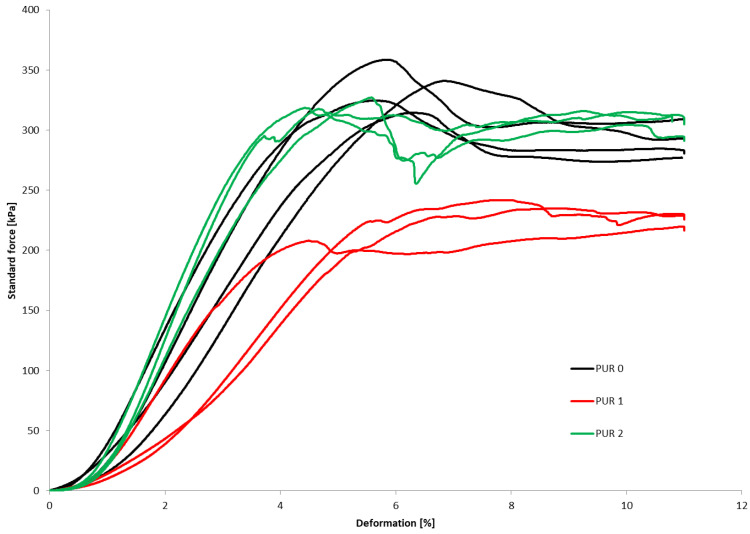
Compressive behavior of samples PUR 0, PUR 1 and PUR 2.

**Figure 3 polymers-14-04616-f003:**
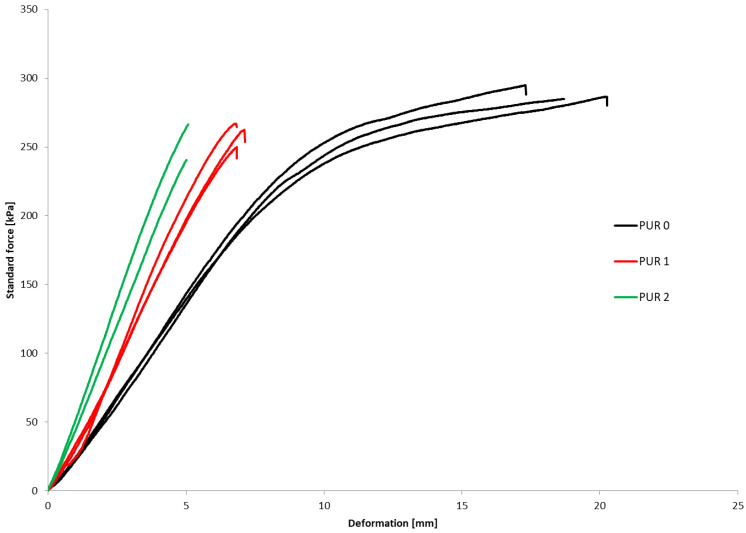
Bending behavior of samples PUR 0, PUR 1 and PUR 2.

**Figure 4 polymers-14-04616-f004:**
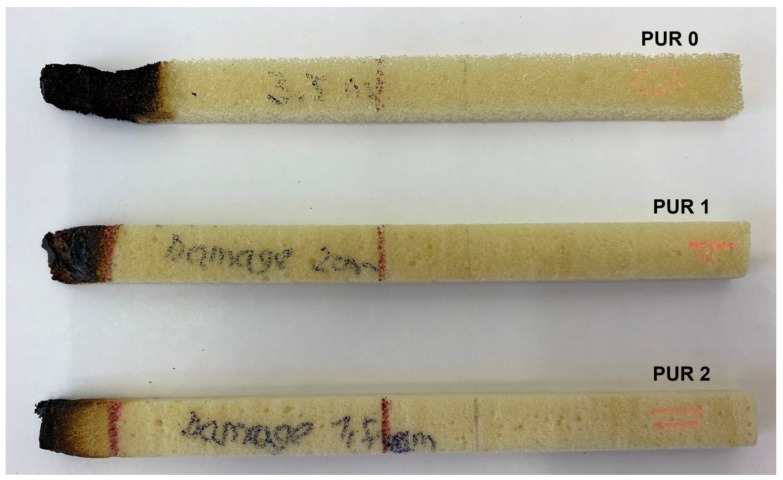
Visual differences between PUR 0, PUR 1, and PUR 2 after the UL-94 HB burning test.

**Figure 5 polymers-14-04616-f005:**
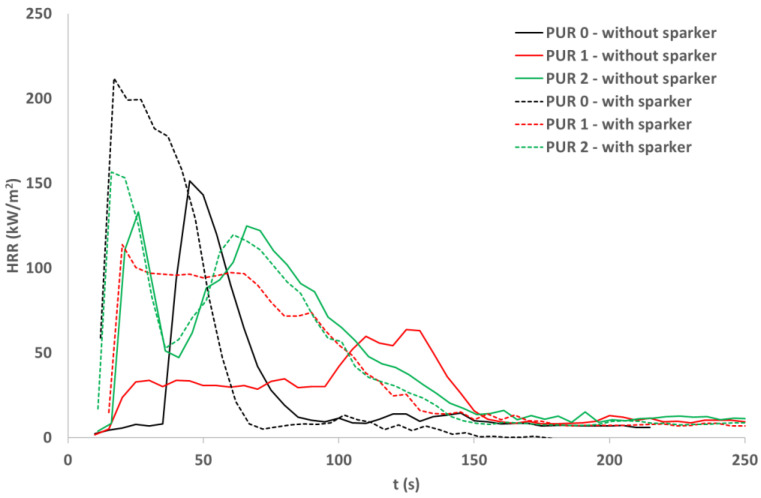
Heat release rate (HRR) of the PURs samples exposed to the 40 kW/m^2^ of heat flux; self-ignition and spark ignition.

**Figure 6 polymers-14-04616-f006:**
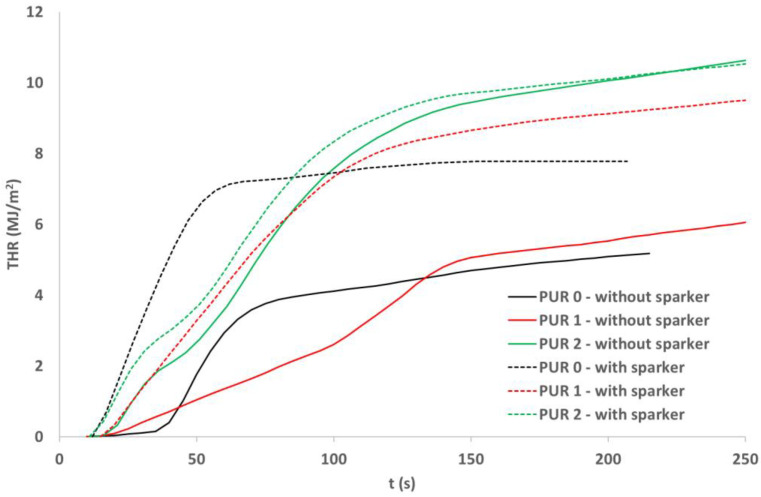
Total heat release (THR) of the PURs exposed to the 40 kW/m^2^ of heat flux; self-ignition and spark ignition.

**Figure 7 polymers-14-04616-f007:**
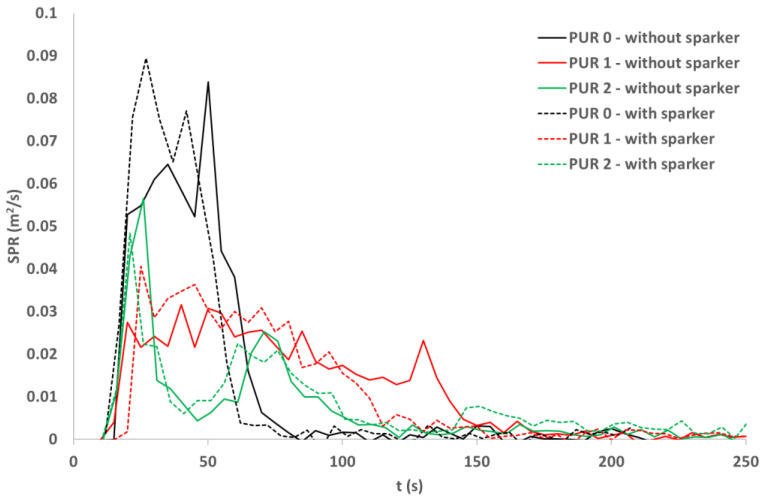
Smoke production rate (SPR) of the PURs samples exposed to the 40 kW/m^2^ of heat flux; self-ignition and spark ignition.

**Figure 8 polymers-14-04616-f008:**
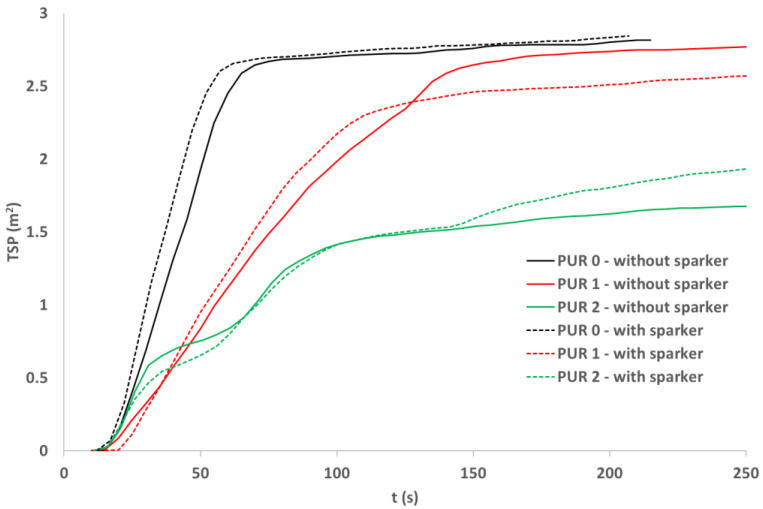
Total smoke production (TSP) of the PURs samples exposed to the 40 kW/m^2^ of heat flux; self-ignition and spark ignition.

**Figure 9 polymers-14-04616-f009:**
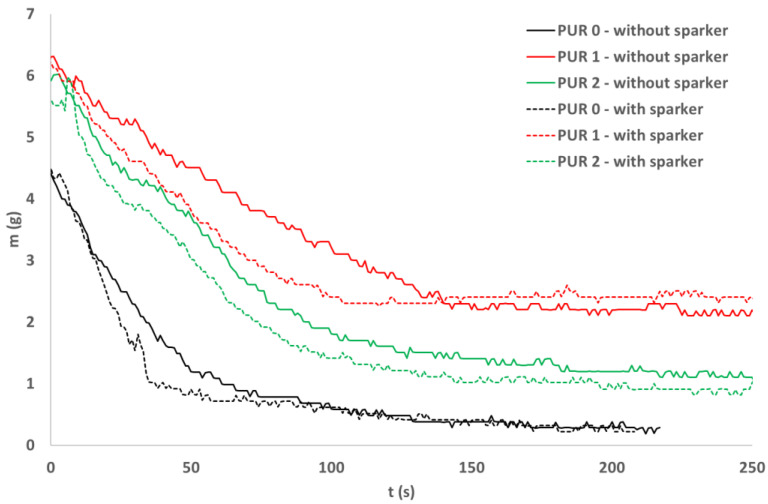
Mass loss of PUR samples exposed to the 40 kW/m^2^ heat flow; self-ignition and spark ignition.

**Figure 10 polymers-14-04616-f010:**
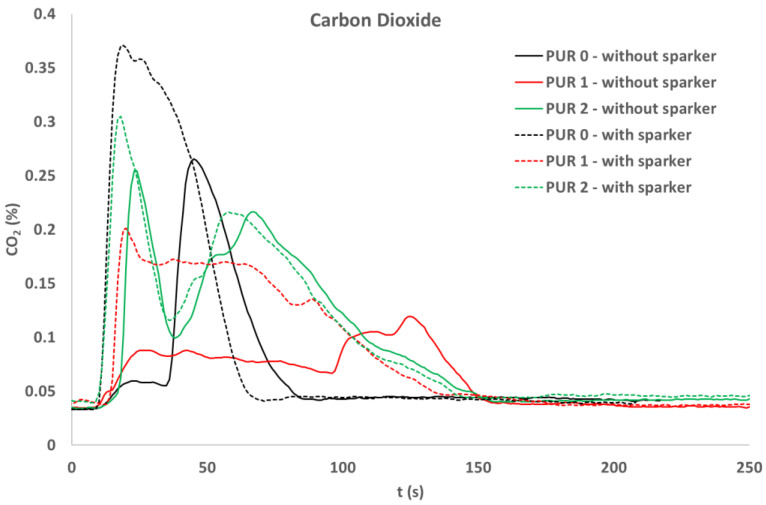
Carbon dioxide release, during exposure of the PURs to the 40 kW/m^2^ of heat flux; self-ignited and ignited with a spark igniter.

**Figure 11 polymers-14-04616-f011:**
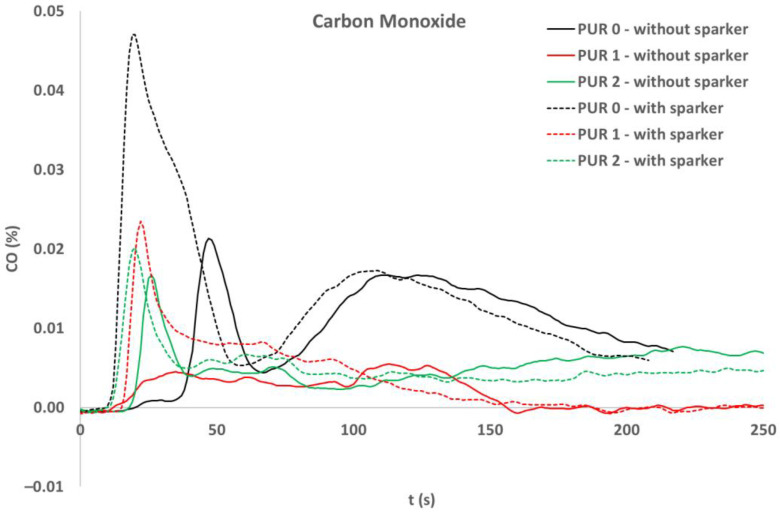
Carbon monoxide release, during exposure of the PURs to the 40 kW/m^2^ of heat flux; self-ignited and ignited with a spark igniter.

**Figure 12 polymers-14-04616-f012:**
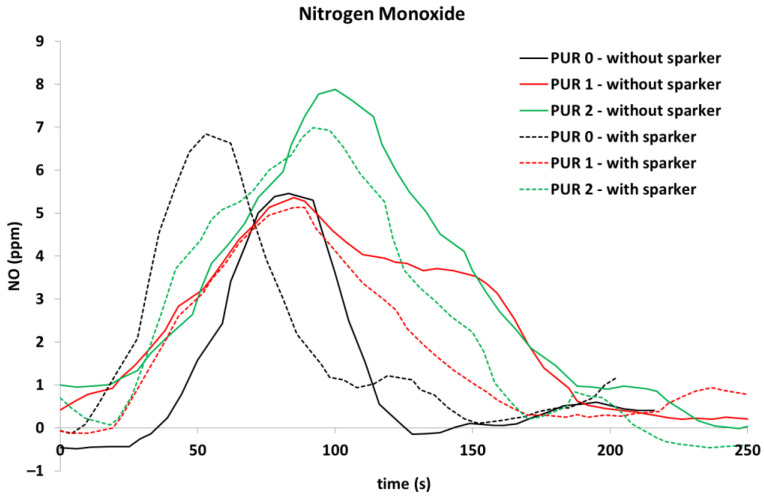
Nitrogen monoxide release, during exposure of the PURs to the 40 kW/m^2^ of heat flux; self-ignited and ignited with a spark igniter.

**Figure 13 polymers-14-04616-f013:**
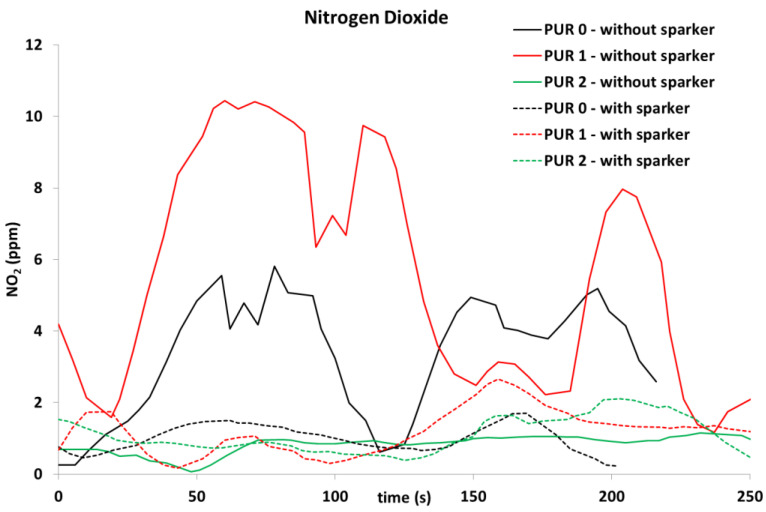
Nitrogen dioxide release, during exposure of the PURs to the 40 kW/m^2^ of heat flux; self-ignited and ignited with a spark igniter.

**Figure 14 polymers-14-04616-f014:**
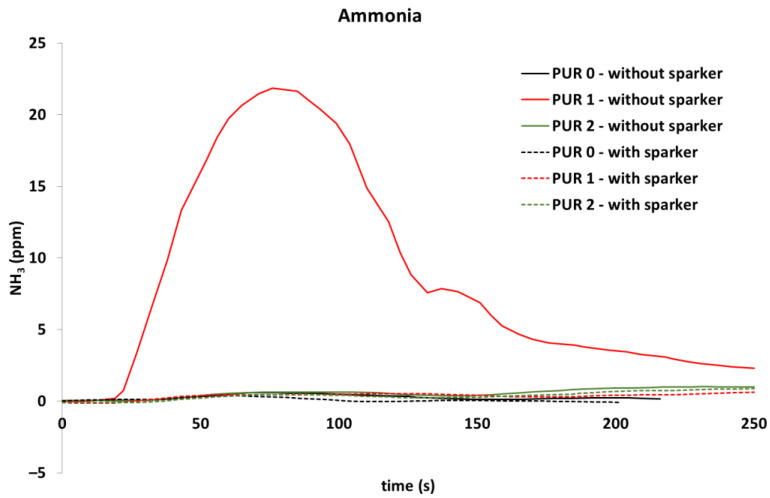
Ammonia release, during exposure of the PURs to the 40 kW/m^2^ of heat flux; self-ignited and ignited with a spark igniter.

**Figure 15 polymers-14-04616-f015:**
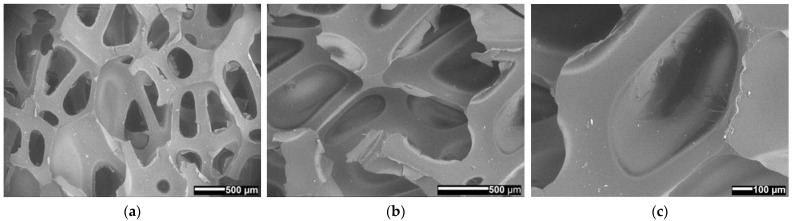
SEM micrographs of the surface of expanded solid polymer PUR 0 sample; at magnification (**a**) 30×, (**b**) 50×, and (**c**) 100×.

**Figure 16 polymers-14-04616-f016:**
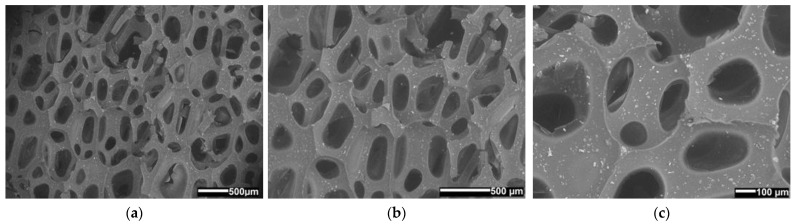
SEM micrographs of the surface of expanded solid polymer PUR 1 sample; at magnification (**a**) 30×, (**b**) 50×, and (**c**) 100×.

**Figure 17 polymers-14-04616-f017:**
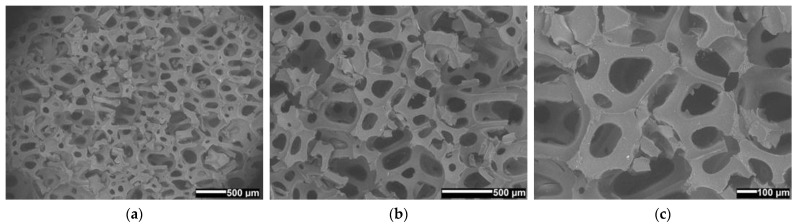
SEM micrographs of the surface of expanded solid polymer PUR 2 sample; at magnification (**a**) 30×, (**b**) 50×, and (**c**) 100×.

**Table 1 polymers-14-04616-t001:** Compositions of the PURs.

Specimens	Raw Material Ratio (%)	Component A (g)	Component B (g)	APP (g)	TATA (g)
PUR 0	Comp. A:Comp. B = 45:55	32.72	40	/	/
PUR 1	Comp. A:Comp. B:APP = 34.61:42.31:23.08	32.72	40	21.82	/
PUR 2	Comp. A:Comp. B:TATA = 34.61:42.31:23.08	32.72	40	/	21.82

**Table 2 polymers-14-04616-t002:** Mechanical properties of PURs.

Specimens	σ_M_ (MPa)	σ_b_ (MPa)
PUR 0	335 ± 19	293 ± 6
PUR 1	220 ± 11	260 ± 9
PUR 2	314 ± 16	253 ± 19

**Table 3 polymers-14-04616-t003:** Hardness of PURs.

Specimens	Durometer Hardness (Shore A)
PUR 0	31 ± 4
PUR 1	16 ± 4
PUR 2	14 ± 3

**Table 4 polymers-14-04616-t004:** Burnt PURs samples in accordance with UL94 standard.

Specimens	V (mm/min)	L (mm)	t (s)	Flame Passed 25 mm Mark	Flame Passed 100 mm Mark	Flammability
PUR 0	35	35	60	Yes	No	Slowly self-extinguishing after withdrawal of fire
PUR 1	20	20	60	No	No	Fast self-extinguishing after withdrawal of fire
PUR 2	15	15	60	No	No	Fast self-extinguishing after withdrawal of fire

V is linear burning rate in mm/minute (mm/min); L is the damaged length, in millimeters (mm); t is time, in seconds (s).

**Table 5 polymers-14-04616-t005:** Prepared test specimens of the PURs, before, during and after exposure to the 40 kW/m^2^ of heat flux in cone calorimeter.

	PUR 0	PUR 1	PUR 2
Before testing	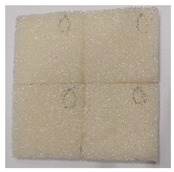	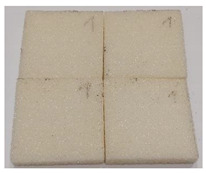	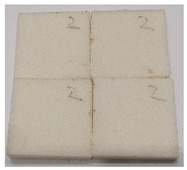
During heat exposure, at the end of flaming	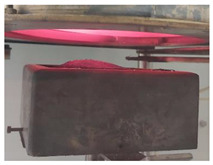	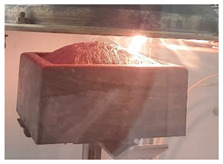	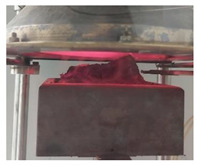
After testing	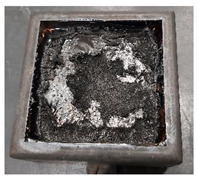	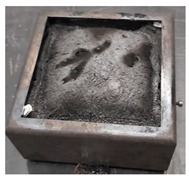	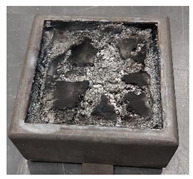

**Table 6 polymers-14-04616-t006:** Ignition times and flame duration for the three PURs exposed to 40 kW/m^2^ of heat flux self-ignited or ignited with a spark igniter.

Specimens	Ignition	Ignition Time (s)	Flame Duration (s)
PUR 0	Self-ignition	33	48
	Ignition with sparks	3	57
PUR 1	Self-ignition	10	150
	Ignition with sparks	3	165
PUR 2	Self-ignition	12	130
	Ignition with sparks	6	134

**Table 7 polymers-14-04616-t007:** Initial mass and loss on ignition for PURs exposed to the 40 kW/m^2^ of heat flux; self-ignited and ignited with a spark igniter.

Specimens	Ignition	Initial Mass (g)	LOI (%)
PUR 0	Self-ignition	4.4	94
	Ignition with sparks	4.5	95
PUR 1	Self-ignition	6.3	67
	Ignition with sparks	6.1	61
PUR 2	Self-ignition	5.92	81
	Ignition with sparks	5.61	83

## Data Availability

Not applicable.

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
