# Peer review of "Flame Retardant Behaviour and Physical-Mechanical Properties of Polymer Synergistic Systems in Rigid Polyurethane Foams"

_polymers, 2022, doi:10.3390/polym14214616_

Round 1

Reviewer 1 Report

The paper entitled “Flame Retardant Behaviour and Physical-mechanical Properties of Polymer Synergistic Systems in Rigid Polyurethane Foams” described the flame retardancy of the PU foams. I think this topic will attract the interest of the readers to some extent. But it has many flaws. I think 3 samples are far not enough. The authors should add more experiment data and compare more samples with each other. The detail comments are as follows.

1.       Please rewrite the abstract part. It is abstract rather than introduction. The Line 7-12 should be rewritten.

2.       Line 16, “do not” should be did not.

3.       The abstract part did not give useful experimental data and simple conclusions. Please rewrite the whole abstract carefully.

4.       The references are too little. Please rewrite the introduction part and summarize others’ work and describe the research background. I think the authors missed some important references, such as 10.1016/j.jmst.2021.05.060, 10.1016/j.compositesa.2021.106486.

5.       How do you choose the weight ratio of each component in Table 1? Can you explain that? I think 3 samples are not enough to discuss the multiply components.

6.       Figure 1 should be deleted.

7.       DTG curves from Figure 2 should be given.

8.       Please combine figure 3-5 together.

9.       How about tensile test results?

10.   The photographs of the samples after UL-94 should be given.

11.   The scale bars are not clearly in Figure 10-11.

Author Response

Dear Sir/Madam.

Please see the attachment (response to the reviewer's comment).

Thank you.

Reviewer 2 Report

In the manuscript, the authors investigated flame retardant behavior and physical-mechanical properties of polymer synergistic systems in rigid polyurethane foams. However, there are some significant issues in the manuscript that must be addressed.

1. In the introduction, the story is vague, with no flow and direction. Again no stress is placed on the gap in previous works which may be filled in by this work.

2. Make sure all abbreviations are written out in full the first time used.

3. Why the char residues of PUR2 is lower than that of pristine PUR?

4. UL-94 is not enough to investigate the flame retardancy of polymers, LOI and cone calorimetry testing must be provided.

5. What is the flame-retardant mechanism of the PUR composites? The corresponding part must be added in the manuscript. Besides, to better clarify the flame-retardant mechanism of PUR composites, more characterization of the char residues must be provided, such as, Raman, FTIR, etc.

Author Response

(The authors gave the same response as above.)

Round 2

Reviewer 1 Report

The authors have answered my comments well. I think it can be published on this journal.

Reviewer 2 Report

It can be accepted.